# Strain Compensation and Trade-Off Design Result in Exciton Emission at 306 nm from AlGaN LEDs at Temperatures up to 368 K

**DOI:** 10.3390/ma14216699

**Published:** 2021-11-07

**Authors:** Shih-Ming Huang, Mu-Jen Lai, Rui-Sen Liu, Tsung-Yen Liu, Ray-Ming Lin

**Affiliations:** 1Department of Radiation Oncology, Chang Gung Memorial Hospital, Keelung 20401, Taiwan; skiwalkergg@gmail.com; 2Jiangxi Litkconn Academy of Optical Research, Longnan City 341700, China; 15007070418@139.com (M.-J.L.); asen2007@126.com (R.-S.L.); 3Department of Electronic Engineering and Institute of Electronics Engineering, Chang Gung University, Taoyuan 33302, Taiwan; 4Department of Radiation Oncology, Chang Gung Memorial Hospital, Linkou 33305, Taiwan

**Keywords:** AlGaN, ultraviolet, light emitting diodes, exciton emission, MOCVD

## Abstract

In this study, we suppressed the parasitic emission caused by electron overflow found in typical ultraviolet B (UVB) and ultraviolet C (UVC) light-emitting diodes (LEDs). The modulation of the p-layer structure and aluminum composition as well as a trade-off in the structure to ensure strain compensation allowed us to increase the p-AlGaN doping efficiency and hole numbers in the p-neutral region. This approach led to greater matching of the electron and hole numbers in the UVB and UVC emission quantum wells. Our UVB LED (sample A) exhibited clear exciton emission, with its peak near 306 nm, and a band-to-band emission at 303 nm. The relative intensity of the exciton emission of sample A decreased as a result of the thermal energy effect of the temperature increase. Nevertheless, sample A displayed its exciton emission at temperatures of up to 368 K. In contrast, our corresponding UVC LED (sample B) only exhibited a Gaussian peak emission at a wavelength of approximately 272 nm.

## 1. Introduction

Compact high-efficiency solid-state ultraviolet (UV) light sources, including light-emitting diodes (LEDs) and laser diodes, are of considerable technological interest for use as alternatives to large, toxic, low-efficiency gas lasers and mercury lamps. In addition, mercury lamps are common sources of UV radiation, particularly at shorter wavelengths, and they are bulky and require high voltages for operation. LEDs, on the other hand, are much smaller and can be operated at lower voltages [1]. UV light can be divided into several bands depending on the application, or industry concerned. For health and safety purposes, UV light is commonly divided into UVA (400–320 nm), UVB (320–280 nm) and UVC (280–100 nm) bands [2,3]. UV light can damage biological systems, particularly at shorter wavelengths; indeed, many sterilization and food-processing systems use UVC light. In contrast, small amounts of UVB light can be therapeutic for certain skin conditions (e.g., psoriasis; optimum wavelength: ca. 311 nm) [4]. In this paper, we report our attempts to use electron/hole pairs bound in excitons as a light source for AlGaN-based LEDs. Introducing this emission resulting from an exciton transition provides exciton-related solid-state light-emitting devices as potential replacements for gas light sources.

In AlGaN, a large exciton binding energy induces strong excitonic effects. For example, the optical absorption spectrum of AlGaN exhibits sharp resonance features [5,6,7] due to dominating direct transitions between the valence and conduction bands at the same K (wave vector) points [8,9,10]. These strong excitonic effects result in a significant transfer of oscillator strength from the band-to-band transition to the 1 s exciton state [11]. The ratio of the oscillator strength of the 1 s exciton state to the band-to-band transitions can reach up to 100-fold [12,13,14]. The AlN (exciton binding energy: 80 meV) is not the only material in which excitons have been used as the basis of light-emitting devices. Several other wide-band-gap semiconductors, including diamond (exciton binding energy: ca. 80 meV) and zinc oxide (exciton binding energy: 60 meV), have stable excitons at room temperature [1]. The binding energy for stable excitons must be greater than the equivalent thermal energy at room temperature. The dominant forces that stabilize excitons are coulombic interactions between the negative and positive charge carriers. There are many factors restricting electrons and holes from forming excitons. Ideally, the K values of the electrons and holes should be the same—as should their velocities. Thus, excitons can only form at the bottom of the conduction band and at the top of the valence band. Nevertheless, if there are too many electrons nearby, they will interfere with the electron/hole coulombic attraction required to form the exciton. In general, exciton emissions are observed from relatively pure materials, not from highly doped materials [8].

The exciton binding energy is a measure of the stability of the electron/hole pair. An exciton having a high binding energy is less likely to be scattered by lattice vibrations to form free charge carriers or undergo conversion to free charge carriers as a result of the presence of impurities or defects. Consequently, the lifetime of an exciton having a high binding energy will be longer than that of one with low binding energy [15,16]. The higher exciton binding energy also increases the radiative rate and decreases the nonradiative rate [17]. These attributes are important to consider when developing photonic devices. The biggest problem faced when preparing nitride-based UVC and UVB LEDs is the ability to incorporate AlN with a high Al content because the doping activation energy of Mg is very high and the doping efficiency is very low. The thickness of the p-AlGaN layer must therefore be increased to satisfy the requirements of quasi-charge-neutrality. To date, however, the conductivity of the neutral region of AlGaN deep ultraviolet (DUV) has not met the conditions of quasi-charge-neutrality when injected into quantum wells (excess electrons are injected and then the same number of holes are injected). Such systems require a longer thermal equilibrium reaction time as well as a larger series resistance, thereby resulting in bias loss [18,19]. In addition, because of the effects of light absorption and the series resistance, the thickness of the p-GaN contact layer cannot be greater than 40 nm. Therefore, the mismatch in the degrees of the electron/hole injection in the active layer of the quantum well will lead to a lack of quasi-charge-neutrality. Some attempts have been made at band engineering in the design of epitaxially structured UV LEDs displaying improved device performance [20,21]. The traditional UVB structure features an electron-blocking layer (EBL) inserted after the quantum well active layer, but it blocks not only the overflow of electrons but also the injection of holes. The possibility of exciton emission can greatly improve the luminous efficiency of an LED device. The exciton binding energies of AlN, diamond and ZnO are 80, 80 and 60 meV, respectively; however, at present, these remains difficult to form quality p–n junctions from these three materials. AlGaN is the most promising material for forming p–n junctions to exploit exciton emissions. Furthermore, the AlN template and sapphire were the best substrates for UVC and UVB emissions. In addition to problems with doping and activation energies, limitations in heterojunction lattice mismatch during epitaxial growth have made it more difficult to improve the luminous efficiency in the UVB band than in the UVC band.

If too many electrons or holes accumulate near the lowest energy of the quantum well, a screening effect would likely break apart the formed exciton pairs. Using the present design and arrangement, we observed exciton emission behavior from AlGaN-based LEDs. With the semiconductor exciton light-emitting device, it was possible to combine electrons and holes in a favorable manner. Because of coulombic attraction, the energy of the electron/hole pair as a whole was increased. Furthermore, because the electrons and holes were close together and moving at the same velocity, the rate of direct recombination was increased, thereby improving the efficiency of the exciton emission. In this study, we obtained AlGaN UVB exciton emission by increasing the number of holes. In addition, we made some sacrifices in terms of the energy and number of photons absorbed in exchange for developing a global green technology.

## 2. Materials and Methods

The epitaxial growth of AlGaN-based LEDs was performed using low-pressure metal–organic chemical vapor deposition (LP-MOCVD). Trimethylaluminum, trimethylgallium, silane, bis(cyclopentadienyl)magnesium and ammonia were used as Al, Ga, Si, Mg and N sources, respectively.

To obtain the UVB LED (sample A), a 2.2 μm-thick AlN buffer layer was first grown on a 2-inch (0001)-oriented sapphire substrate. Then, an interlayer consisting of 30 periods of a 2.5 nm AlN/17.5 nm AlGaN superlattice, with the same Al composition (0.72), was grown on the AlN buffer layer. A 1.5 μm-thick undoped Al_0.6_Ga_0.4_N layer was then grown on the superlattice interlayer. Subsequently, a 2 μm-thick Si-doped n-Al_0.5_Ga_0.5_N layer was grown as an n-neutral region and n-contact layer. The active region included four periods of Al_0.35_Ga_0.65_N (2 nm)/Al_0.45_Ga_0.55_N (8 nm) MQWs. A two-fold Mg-doped p-Al_0.55_Ga_0.45_N (10 nm)/Mg-doped p-Al_0.4_Ga_0.6_N (2 nm) structure, replacing the conventional AlGaN EBL, was grown, followed by a 50 nm-thick Mg-doped p-Al_0.3_Ga_0.7_N–to–p-GaN grading layer. Finally, a 20 nm-thick Mg-doped p-GaN layer was deposited to serve as the p-contact layer (Figure 1). To facilitate exciton emissions of nitride-based compounds in the UVB band at room temperature, it will be necessary to shorten the relaxation time of the holes in the neutral region. Accordingly, by considering the requirements for light absorption and the relaxation time of the holes, in this study, we designed an LED containing a 20 nm-thick layer of p-GaN. Prior to depositing the p-GaN layer, we abandoned the bulk AlGaN EBL and replaced it with a two-fold p-Al_0.55_GaN/p-Al_0.4_GaN structure to improve the crystal quality. After depositing the two-fold layer structure, we wished to minimize the two-dimensional electron gas (2DEG) produced by the spontaneous polarization and piezoelectric polarization between materials, potentially leading to electron/hole recombination and parasitic luminescence in the neutral region of the p-type structure; thus, we also deposited a 50 nm-thick p-AlGaN–to–p-GaN grading composition layer. Finally, we deposited the 20 nm-thick p-GaN layer to ensure good ohmic contact. When cubic crystals are subjected to shear stress and dilatation stress, they can experience tensile or compressive strain. Therefore, we introduced a trade-off in the design of the layer structure to ensure strain compensation. We expected the n-AlGaN layer to induce a compressive strain to the multiple quantum wells (MQWs) of the active layer, while the effective p-AlGaN neutral region would induce a tensile strain. Therefore, the structure of the UVB device would not only depress the piezoelectric polarization effect as a whole but also control the strain-relaxed process under plastic deformation, thereby maintaining the epitaxial quality of the device.

To obtain the UVC LED (sample B), the AlN buffer layer, 30-period superlattice and undoped Al_0.6_Ga_0.4_N layer were prepared in the same manner as described above for sample A. Subsequently, a 2 μm-thick Si-doped n-Al_0.6_Ga_0.4_N layer was grown as the n-contact. The active region included four periods of Al_0.45_Ga_0.55_N (2 nm)/Al_0.55_Ga_0.45_N (8 nm) MQWs. A two-fold Mg-doped p-Al_0.55_Ga_0.45_N (10 nm)/Mg-doped p-Al_0.45_Ga_0.55_N (2 nm) structure was then grown, replacing the conventional AlGaN EBL, followed by a 50 nm-thick Mg-doped p-Al_0.3_Ga_0.7_N–to–p-GaN grading layer. Finally, a 20 nm-thick Mg-doped p-GaN layer was deposited to serve as the p-contact layer (Figure 1).

After performing the LP-MOCVD growth processes, the samples were annealed in an N_2_ ambient to activate the Mg-dopants. The LED chips were fabricated using standard chip-processing technologies. Mesa structures were defined through inductively coupled plasma etching to expose the n-Al_0.5_Ga_0.5_N layer surface. Ti/Al/Ti/Au (100/200/30/100 nm) n-contacts were deposited through electron–beam evaporation and annealed through rapid thermal annealing at 980 °C for 60 s. To form the transparent p-contact, a 50 nm-thick layer of indium tin oxide (ITO) was sputter-deposited and annealed at 600 °C for 10 min. The LED chips were completed with the deposition of Ti/Pt/Au (50/30/100 nm) and AuSn (3 μm); the p-contact area was approximately 0.14 mm^2^.

The prepared flip-chips were bonded on an AlN direct plating ceramic lead frame using the eutectic method (AD211 plus, ASM); a covering quartz glass served as an optical lens. The packaged samples were soldered onto an Al metal core printed circuit board. The light output power (LOP), current–voltage characteristics and electroluminescence (EL) spectra of the samples were measured using an ATA-5000 LED photoelectric measurement system (Everfine) equipped with a 30 cm-diameter integrating sphere. During the measurement, the temperature of the heat-sink mounting the packaged sample was controlled under 298, 323, 348, or 368 K at a driving direct current of 40 mA (current density: ca. 28.5 A/cm^2^). To minimize the effect of the self-heating of the chips, the stop interval was set to 3 min during each continuous wave (CW) measurement.

## 3. Results and Discussion

Figure 2a,b present the EL band peak positions (measured at room temperature) of the two samples before and after performing the burning tests at 40 mA. Sample A exhibited a clear exciton emission, with its peak near 306 nm, and a band-to-band emission at 303 nm. The peak at 306 nm might have also been due to the quantum well composition causing spatial potential fluctuations. We observed, however, a full width at half maximum (FWHM) of less than 10 nm. Compositional fluctuations would result in larger FWHMs; therefore, we examined whether this peak primarily resulted from the exciton emission [22]. Because the excitons existed in the crystal and not in free space, the binding energy would decrease as a result of the effects of the reduced mass and the dielectric constant, as implied by Equation (1), where *m*_r_ is the reduced mass, *R_y_* is the Rydberg energy (equal to 13.6 eV), *ε_r_* is the relative dielectric constant and *n* is an integer (≥1) [8].
(1)$$E_{x}=-\left(\frac{m_{r}q^{4}}{2h^{2}\epsilon^{2}}\right)\frac{1}{n^{2}}=-\frac{\left(m_{r}/m_{0}\right)}{\epsilon_{r}^{2}}\frac{R_{y}}{n^{2}}$$

After applying Equation (1), we obtained an exciton bonding energy for Al_0.35_Ga_0.65_N of approximately 39 meV. Accordingly, we confirmed that the signal near 306 nm arose from exciton emission. Sample B provided only a Gaussian peak emission at a wavelength of approximately 272 nm. We calculated the Al content in the AlGaN structure by using the Schrödinger wave equation with a quantized level shift and Vegard’s law; the Al_0.45_Ga_0.55_N band-to-band emission was approximately 270 nm. Therefore, we mainly attribute the emission at 272 nm to the band-to-band transition. For both samples, increasing the temperature caused the intensity of the band-to-band emission to decrease and shift to a slightly longer wavelength. The relative intensity of the exciton emission in sample A decreased as a result of the thermal energy effect. Sample A provided the exciton emission at temperatures up to 368 K. Several reports of UVC LEDs have described the presence of parasitic peaks extending from the low-energy side of the main peak [23]. These parasitic peaks are presumably associated with recombination occurring through deep level defects and electron overflow and through polarization doping recombination in the p-AlGaN neutral region. In this present study, we did not observe any obvious parasitic peaks from sample B at any temperature up to 368 K. Therefore, we infer that defect-related recombination did not occur in sample B under the injection conditions in the temperature range from approximately 298 to 368 K. Santi et al. reported [24] that the normalized intensity of parasitic peak 3 (ca. 340 nm) from a UVB LED monotonically decreased upon increasing the temperature from 100 to 400 K; they suggested that, because of its broad shape, this peak originated from radiative transitions through deep levels in the quantum barrier next to the EBL.

Figure 3a,b reveal the temperature-dependence of the peak wavelengths of samples A and B at 40 mA. The peak wavelengths (and FWHMs) of both samples slightly shifted—from 306.0 (9.612) to 306.6 (11.02) nm for sample A and from 271.8 (10.34) to 272.5 (11.24) nm for sample B—upon increasing the temperature from 298 to 368 K. Consistently with Varshni’s law, these red-shifts were caused by thermal expansion upon increasing the temperature [25]. At temperatures of up to 368 K, the emission of sample A appeared as two Gaussian peaks: the exciton emission and the band-to-band emission.

To obtain a deeper understanding of the temperature-dependence of the emissions from the UVB and UVC LEDs, we measured the non-normalized light output power of samples A and B at various temperatures (Figure 4). Upon increasing the temperature, the light output power from both samples monotonically decreased; however, the slope of the line for sample A was much lower than that for sample B. The exciton emission from sample A was less likely to be scattered by phonons. The temperature-reliability of a photonic device is affected by the carrier confinement factor, with a larger discontinuity in the conductive band enhancing the thermal stability. Because the values of Δ*E_c_* of samples A and B were nearly identical, the electroluminescence of sample A must have arisen, in part, from the exciton emission. The characteristic temperature (*T*_0_) can be derived from fitting the characteristic temperature equation of a semiconductor laser. According to the fitting data derived from the temperature-dependence of the light output power, the values of *T*_0_ of samples A and B were 328 and 108 K, respectively. Chhajed et al. reported that a larger value of *T*_0_ could decrease the contribution of non-radiative recombination to the total carrier recombination due to the saturation of the non-radiative recombination paths; as a result, the activation energy of the non-radiative recombination centers and the energy required to overcome the confining potentials would both increase [26].

Ploch et al. [27] investigated the influence of the dislocation density and the barrier height on the temperature stability of the light output power from a 380 nm LED. The decrease in the temperature-stability was more pronounced in a sample having a high threading dislocation density (TDD) and it was weakly influenced by the barrier height when the current density was less than 50 A/cm^2^. In this present study, both of our samples had the same structure below the Si-doped n-contact layer, and no indium was present in the MQW active region. Nevertheless, sample A was more temperature-insensitive than sample B in terms of its light output power at a current density of approximately 28.5 A/cm^2^. Hence, we attribute the improved temperature stability of sample A to its lower effective TDD and superior carrier confinement in the active region, due to the exciton quantum confinement effect. The behavior of sample A can also be explained by considering the fact that its exciton emission was less likely to be scattered by lattice vibrations to form free charge carriers.

## 4. Conclusions

We developed a new approach towards a nitride-based UVB LED that realizes an exciton emission in the UVB band. Our design, based on strain-compensation, suppressed the parasitic emission resulting from electron overflow that is found in typical UVB devices. Despite the content of Al in the luminescent layer of sample A being smaller than that of sample B, sample A was more temperature-insensitive in its light output power at a current density of approximately 28.5 A/cm^2^. According to the fitting of the temperature dependence of the light output power, the characteristic temperatures (*T*_0_) of samples A and B were 328 and 108 K, respectively.

## Figures and Tables

**Figure 1 materials-14-06699-f001:**
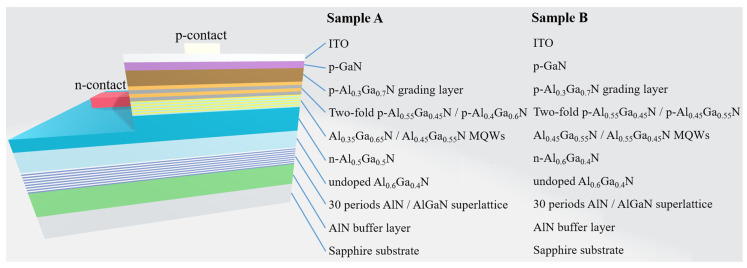
Schematic representation of the epitaxial structures of samples A and B.

**Figure 2 materials-14-06699-f002:**
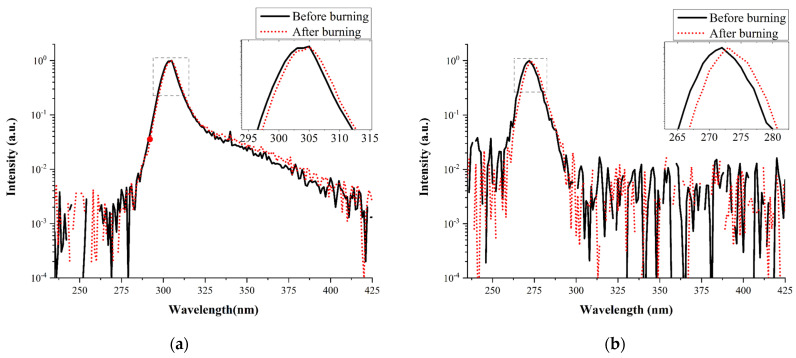
EL band peak positions measured at room temperature before and after the burning tests of (**a**) sample A (*V*_f_ = 5.246 V @40 mA) and (**b**) sample B (*V*_f_ = 5.530 V @40 mA). The inset is the enlargement of the peak portion.

**Figure 3 materials-14-06699-f003:**
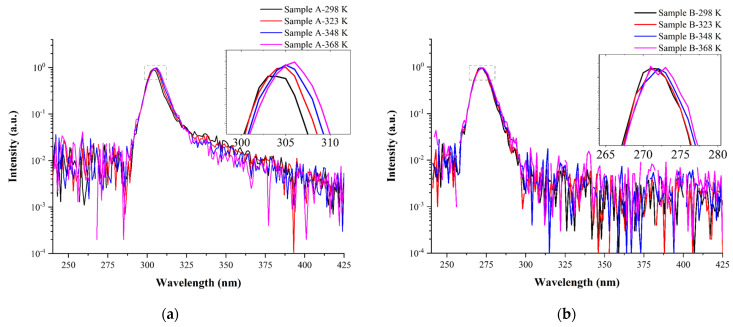
Temperature-dependence of the EL peak wavelengths of (**a**) sample A (*V*_f_ = 5.246 V @40 mA) and (**b**) sample B (*V*_f_ = 5.530 V @40 mA). The inset is the enlargement of the peak portion.

**Figure 4 materials-14-06699-f004:**
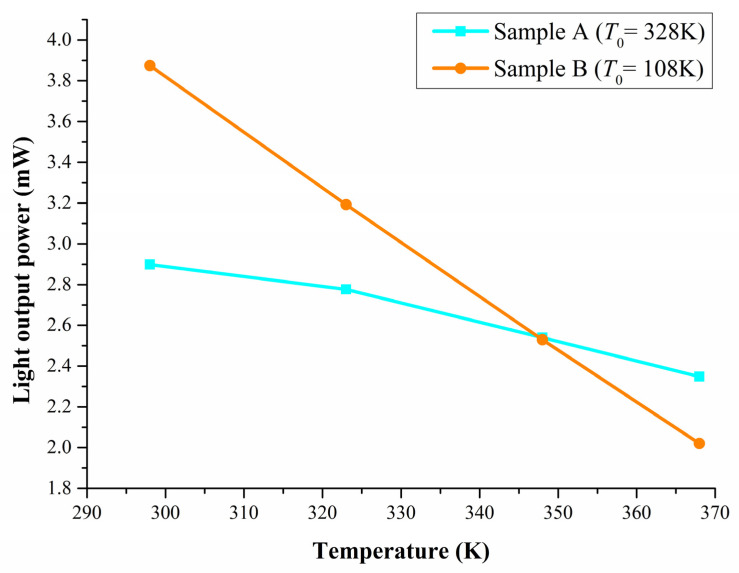
Light output power from samples A and B measured at various temperatures at 40 mA.

## Data Availability

All data contained within the article.

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
