# Peer review of "Strain Compensation and Trade-Off Design Result in Exciton Emission at 306 nm from AlGaN LEDs at Temperatures up to 368 K"

_materials, 2021, doi:10.3390/ma14216699_

Round 1

Reviewer 1 Report

This work demonstrates two LEDs with strain compensation and trade-off design, and both EL and TDEL (298~363K) test was carried out. A major revision is required before considering for publication in materials.

Below are few comments.

  1. As “strain compensation and trade-off design” is a highlight of the proposed LEDs in this manuscript, more detailed discussion and characterization about strain engineering of the proposed LEDs should be carried out.

  1. In Figure 1, The structure diagram of sample A is not clear enough, there is no electrode information, and there is no structure diagram of sample B. It is better to have a comparison diagram of sample A and sample B.

  1. Figure 2 and Figure 3 have no labels (a) and (b), and their resolution is too low. Also, the drawing is not consistent between figures, for example, the tick marks in x-axis of both figures should be the same.

  1. Figure a and b in Figure 2 have different abscissa ranges. Figure 2b should expand the abscissa range to prove that there are indeed no parasitic peaks as described in the manuscript.

  1. It can be seen from Figure 4 that the LOP of sample A is smaller than that of sample B when the temperature is less than 350K. It shows that although Sample A is not sensitive to temperature, the LOP is greatly reduced which could hurt the overall performance of the device, right?

  1. Besides, there is no I-V, LEE or EQE data in the manuscript. Please provide them as much as you can so we can understand and evaluate their performance better. For example, on page 4, line 174, the light output power (LOP), the current–voltage characteristics, and the electroluminescence (EL) spectrum of the samples were measured…” However, no current–voltage characteristics were shown in the manuscript.

  1. In Fig. 2 (b), sample B shows a redshift in peak wavelength after burning test, while sample A shows no change in peak wavelength. Please explain it.

  1. In Fig. 3, it is hard to observe the redshift of both samples, please enlarge the emission peak for better readability.

  • Most of the references listed in the manuscript are too old, and does not include the recent progress in the development of UVLEDs, newly  references should be added:  Optics Letters Vol. 46, Issue 13, pp. 3271-3274 (2021).

  1. The paper also missed lots of important references in the main content, as listed below:

(1) On page 1, line 35, the statement “On the other hand, small amounts of UVB can be therapeutic for certain skin conditions” lacks related reference.

(2) On page 2, line 60, the statement “An exciton with high binding energy … an exciton having a high binding energy will have a lifetime longer than that of one with a weak binding energy” lacks related reference.

(3) On page 4, line 191, the statement “composition fluctuations will result in a larger FWHM” lacks related reference. See Advanced Functional Materials, 29(48), 1905445, 2019

(4) On page 5, line 228, the statement “Consistent with Varshni’s law, the red-shifts were 228 caused by thermal expansion with increasing the temperature” lacks related reference.

(5) On page 5, the reference of equation 1 is lacked, and the physical significance of each symbol should be explained clearly for better understanding.

There are some errors in the paper, see the examples below:

  1. On page 2, line 95, “And also make the series resistance larger, resulting in bias loss.” The sentence lacks the subject.

  1. On page 3, line 136, “In this paper, Al-GaN UVB exciton emission by increasing the numbers of Holes.” This sentence is incomplete.

  1. On page 4, line 192, “so we have checked this peak primarily was caused by exciton emission. ”

Reviewer 2 Report

See attached file

Round 2

Reviewer 1 Report

the revision can be accepted now. 

Author Response

This reviewer had no additional questions.

Reviewer 2 Report

see attachment
